# Intraoperative Intercostal Nerve Block for Postoperative Pain Control in Pre-Pectoral versus Subpectoral Direct-To-Implant Breast Reconstruction: A Retrospective Study

**DOI:** 10.3390/medicina56070325

**Published:** 2020-06-30

**Authors:** Jin-Woo Park, Jeong Hoon Kim, Kyong-Je Woo

**Affiliations:** Department of Plastic and Reconstructive Surgery, Ewha Womans University Mokdong Hospital, College of Medicine, Ewha Womans University, Seoul 07985, Korea; burnscar@naver.com (J.-W.P.); kimsbrothers@hanmail.net (J.H.K.)

**Keywords:** intercostal nerve block, postoperative pain, pain control, direct-to-implant breast reconstruction, prosthesis, implant

## Abstract

*Background and Objectives:* Patients undergoing mastectomy and implant-based breast reconstruction have significant acute postsurgical pain. The purpose of this study was to examine the efficacy of intercostal nerve blocks (ICNBs) for reducing pain after direct-to-implant (DTI) breast reconstruction. *Materials and Methods:* Between January 2019 and March 2020, patients who underwent immediate DTI breast reconstruction were included in this study. The patients were divided into the ICNB or control group. In the ICNB group, 4 cc of 0.2% ropivacaine was injected intraoperatively to the second, third, fourth, and fifth intercostal spaces just before implant insertion. The daily average and maximum visual analogue scale (VAS) scores were recorded by the patient from operative day to postoperative day (POD) seven. Pain scores were compared between the ICNB and control groups and analyzed according to the insertion plane of implants. *Results*: A total of 67 patients with a mean age of 47.9 years were included; 31 patients received ICNBs and 36 patients did not receive ICNBs. There were no complications related to ICNBs reported. The ICNB group showed a significantly lower median with an average VAS score on the operative day (4 versus 6, *p* = 0.047), lower maximum VAS scores on the operative day (5 versus 7.5, *p* = 0.030), and POD 1 (4 versus 6, *p* = 0.030) as compared with the control group. Among patients who underwent subpectoral reconstruction, the ICNB group showed a significantly lower median with an average VAS score on the operative day (4 versus 7, *p* = 0.005), lower maximum VAS scores on the operative day (4.5 versus 8, *p* = 0.004), and POD 1 (4 versus 6, *p* = 0.009), whereas no significant differences were observed among those who underwent pre-pectoral reconstruction. *Conclusions:* Intraoperative ICNBs can effectively reduce immediate postoperative pain in subpectoral DTI breast reconstruction; however, it may not be effective in pre-pectoral DTI reconstruction.

## 1. Introduction

The adequate management of immediate postoperative pain after breast surgery is important for improving patients’ well-being in the immediate postoperative period and for reducing pain-induced restriction of movement predisposing patients to poor recovery [1]. Inadequate management of pain in the immediate postoperative period affects patients’ life quality and can also have severe consequences such as chronic postoperative pain [2,3,4,5]. The incidence of chronic postoperative pain after mastectomy and breast reconstruction has been reported in up to 50% of patients [6,7,8].

The management of immediate postoperative pain includes the use of intravenous analgesics, botulinum toxin, glucocorticoids, muscle relaxers, nonsteroidal agents, indwelling pain catheters, paravertebral nerve blocks, and intercostal nerve blocks (ICNBs) [1,4]. Among them, ICNBs were introduced in mastectomy by McCann [9] as an adjunctive to general anesthesia, and the use of ICNBs has been reported to be a safe and effective method to manage immediate postoperative pain in patients undergoing tissue expander reconstruction [1] and augmentation mammoplasty [10,11]. ICNBs can be performed in breast reconstruction under the direct vision of the intercostal spaces intraoperatively by the operator without an anesthesiologist [1].

Direct-to-implant (DTI) breast reconstruction is one of the most recent advances in breast reconstruction and is currently gaining popularity [12,13,14] due to its oncologic safety, cost effectiveness, and aesthetic outcome [15,16,17,18]. Several studies have investigated the techniques and surgical and aesthetic outcomes of DTI breast reconstruction; however, limited studies have examined the use of intraoperative ICNBs with a local anesthetic for patients undergoing DTI breast reconstruction [1]. Particularly, there have been no studies that have evaluated the differences in the natural course and management of pain according to the insertion plane of implants. The main objective of the current study was to evaluate the efficacy of intraoperative nerve blocks using ropivacaine for postoperative pain after DTI breast reconstruction. The secondary objective was to compare pain scores between pre-pectoral and subpectoral DTI breast reconstruction.

## 2. Materials and Methods

### 2.1. Patient Cohort

The study was conducted in accordance with the Declaration of Helsinki, and the protocol was approved by the institutional review board of Ewha Womans University Mokdong Hospital (no. 2020-05-01). Consecutive patients who underwent immediate unilateral DTI reconstruction with an acellular dermal matrix at our institution between January 2019 and March 2020 were retrospectively reviewed. Patients who were given intravenous patient-controlled analgesia (IV PCA) postoperatively were included. Patients who had a previous breast procedure, including breast-conserving surgery for previous malignancy, augmentation mammoplasty, mastopexy, and reduction mammoplasty, and a history of radiation therapy were excluded from this study. A modified Charlson comorbidity index was calculated as a summation of the overall extent of comorbidities for each patient [19,20]. Three experienced oncologic surgeons with over 10 years of experience performed the mastectomies, and the senior author (K.-J.W.) performed all reconstruction procedures.

### 2.2. Surgical Technique

Single-stage breast reconstruction with silicone gel implants was performed in all nipple-sparing mastectomy (NSM) cases and skin-sparing mastectomy (SSM) cases with minimal skin flap excisions. Under a general anesthesia, IV fentanyl (Hana Pharm Co., Ltd., Seoul, Korea) was routinely administered with a dose of 3 µg per kg of the patient’s body weight. For patients who could not afford a large acellular dermal matrix, had a thin mastectomy skin flap (which can cause significant rippling), or had compromised mastectomy skin flap perfusion, the implant was placed in the subpectoral space. An acellular dermal matrix (human cadaveric) was used for inferior and lateral support and implant coverage. Otherwise, implants were placed in pre-pectoral spaces and draped with a 16 × 16 cm or 18 × 18 cm acellular dermal matrix. For patients receiving nerve blocks, 4 cc of 0.2% ropivacaine (Mitsubishi Tanabe Pharma Korea Co., Ltd., Seoul, Korea) was injected into each intercostal space from T2 to T5 just before the placement of the implant. A syringe (10 cc) was connected to a 23-gauge butterfly needle to check and maintain the penetration depth during injection. The needle was angled 15° cephalad, and the penetration depth was 5 mm, which was marked by a sterile tape on the needle to achieve a consistent depth of injection as previously described [21]. Injections were given after a surgeon confirmed that neither blood nor air was aspirated. Injection sites were the most lateral portion of accessible 2nd to 5th intercostal spaces near the mid to anterior axillary line [1,4]. After the placement of the implant, two closed suction drains were placed in the subpectoral and pre-pectoral spaces for subpectoral reconstruction and in the upper and lower pole of the pre-pectoral spaces for pre-pectoral reconstruction. Drains were removed when the drainage amount was less than 30 mL over 24 h for two consecutive days. Patients were discharged after all of the surgical drains were removed. Prophylactic antibiotics were administrated until drain removal.

After the surgery, fentanyl was administered intravenously according to the patient’s vital signs or complaint of pain in the post-anesthesia care unit (PACU). An IV PCA device administering fentanyl and ramosetron (Daiichi Sankyo Korea Co., Ltd., Seoul, Korea) was routinely used after surgery. The IV PCA started with a total volume of 60 mL consisting of 10–12 µg fentanyl per kg of the patient’s body weight and was continuously infused at a rate of 0.5 mL/h. The device was set to deliver 0.5 mL boluses whenever the patient pressed the designated button. Patients were encouraged to press the button whenever they felt pain. The IV PCA device was removed at 48 h postoperatively, and the remaining dose was measured. After the IV PCA device was removed, aceclofenac (100 mg) was administered orally twice daily for pain control. When the patient required additional analgesia, pain control was augmented using IV acetaminophen, IV ketorolac, or oral acetaminophen. For comparison, pain medications were converted to morphine equivalents to calculate the total usage of pain medication. Morphine equivalents administered in the PACU and after the PACU were analyzed.

Patients were educated about visual analogue scale (VAS) scores on the day before the surgery. The daily average and maximum VAS scores were recorded by the patient using a self-recording VAS score sheet, which was given to the patient at the time of patient education. A VAS score of 5 was defined as the degree of pain at which the patient found it difficult to sleep or rest without additional pain control, and a VAS score of 10 was defined as pain as severe as death. Pain degree was recorded from the operative day to postoperative day 7. The VAS score sheets were collected after postoperative day 7 and analyzed. If patients were discharged earlier than postoperative day 7, an oral pain medication was prescribed, and patients were asked to complete the self-recording VAS score sheet and the sheets were retrieved at the first visit to the outpatient clinic. All patients, except two, patients were discharged after 7 postoperative days.

### 2.3. Analysis

Data obtained from the chart review included postoperative PCA usage and patient-reported VAS pain scores from the day of operation to postoperative day 7. The daily average and maximum VAS scores were compared between the pre-pectoral reconstruction and subpectoral reconstruction groups for patients who did not receive ICNBs to analyze the natural course of postoperative pain according to the insertion plane of implants. To evaluate the effect of ICNBs, the daily average and maximum VAS scores were compared between the ICNB and control groups for all patients. The subgroup analyses were performed on the insertion plane of implants to evaluate the difference in the effect of ICNBs between patients who underwent subpectoral reconstruction and those who underwent pre-pectoral reconstruction.

The mean with standard deviation or median with interquartile range were used to summarize continuous variables based on the distribution of the data, and the frequency and proportion were used to describe categorical variables. Shapiro–Wilk tests were conducted to test normal distribution. The clinical and operative variables were compared between the groups using two-sample t-test or Mann–Whitney U test for continuous variables and the Chi-squared test or Fisher’s exact test for categorical variables. The statistical significance was determined by *p* < 0.05. All analyses were performed using SPSS version 23.0 (SPSS Inc., Chicago, IL, USA).

## 3. Results

Among 78 patients who underwent DTI breast reconstruction after NSM or SSM in the study period, 67 patients met the inclusion criteria and were included in this study. A total of 11 patients were excluded as they did not use IV PCA or discontinued using it before 48 h postoperatively due to side effects. There were no immediate nerve block-related complications observed in patients who received intraoperative ICNBs. A total of 33 patients (49.3%) underwent DTI breast reconstruction with pre-pectoral placement of the implant, and the remaining 34 patients (50.7%) had subpectoral implant placement. The mean patient age was 47.91 ± 8.06 years, and the mean body mass index (BMI) was 22.63 ± 2.63 kg/m^2^. There were no significant differences between the pre-pectoral and subpectoral reconstruction groups in age, BMI, Charlson comorbidity index score, smoking history, previous chemotherapy, mastectomy type, axillary lymph node management, and morphine equivalents. The mastectomy weight was significantly higher in the subpectoral group than in the pre-pectoral group (312.3 g versus 236.4 g, *p* = 0.019); however, the implant volume was not significantly different between the two groups (325.7 mL versus 285.9 mL, *p* = 0.103) (Table 1).

The comparison of VAS scores between the pre-pectoral and subpectoral reconstruction groups for patients who did not receive ICNBs is shown in Table 2. On the operative day, the subpectoral reconstruction group showed a higher median with an average VAS score (7 versus 6, *p* = 0.062) with marginal significance. The maximum VAS score (8 versus 6.5, *p* = 0.108) was higher on the operative day; however, the difference did not reach statistical significance. The average and maximum VAS scores were also higher among patients who underwent subpectoral reconstruction than among those who underwent pre-pectoral reconstruction from postoperative day one to day seven; however, the difference was not statistically significant.

### 3.1. Comparison of Pain Scores between the ICNB and Control Groups

Among the selected 67 patients, 31 patients were included in the ICNB group, and the remaining 36 patients were included in the control group. Table 3 shows the comparison of clinical and surgical characteristics between the ICNB and control groups. There were no significant differences between the two groups in age, BMI, Charlson comorbidity index score, smoking history, previous chemotherapy, mastectomy type, mastectomy weight, implant volume, axillary lymph node management, and morphine equivalents. In the comparison of daily VAS scores between the ICNB and control groups, the median of average (4 versus 6, *p* = 0.047) and maximum (5 versus 7.5, *p* = 0.030) VAS scores on the operative day and maximum (4 versus 6, *p* = 0.030) VAS score on postoperative day one were significantly lower in the ICNB group than in the control group (Table 4).

### 3.2. Subpectoral Reconstruction

Among 34 patients who underwent subpectoral reconstruction, 16 patients were included in the ICNB group, and 18 patients were included in the control group. There were no significant differences between the ICNB and control groups in age, BMI, Charlson comorbidity index score, smoking history, previous chemotherapy, mastectomy type, mastectomy weight, implant volume, axillary lymph node management, and morphine equivalents (Table 5). Figure 1 shows the significant differences in the average and maximum VAS scores on the operative day and maximum VAS score on postoperative day one between the ICNB and control groups. On the operative day, the median of the average (4 versus 7, *p* = 0.005) and maximum (4.5 versus 8, *p* = 0.004) VAS scores were significantly lower in the ICNB group than in the control group. On postoperative day one, the median of the average VAS score was lower in the ICNB group than in the control group with marginal significance (3.5 versus 5, *p* = 0.060), and the maximum VAS score was significantly lower in the ICNB group than in the control group (4 versus 6, *p* = 0.009).

### 3.3. Pre-Pectoral Reconstruction

Among 33 patients who underwent pre-pectoral reconstruction, 15 patients were included in the ICNB group, and 18 patients were included in the control group. There were no significant differences between the ICNB and control groups in age, BMI, Charlson comorbidity index score, smoking history, previous chemotherapy, mastectomy type, mastectomy weight, implant volume, axillary lymph node management, and morphine equivalents (Table 6). Figure 2 compares the postoperative average and maximum VAS scores between the ICNB and control groups. Both the average and maximum VAS scores were decreased with each successive postoperative day. On the operative day, the median of the average VAS score was 5 in the ICNB group and 6 in the control group (*p* = 0.891), and the maximum VAS score was 5 in the ICNB group and 6 in the control group (*p* = 0.782). On postoperative day one, the median of the average VAS score was 4 in the ICNB group and 4 in the control group (*p* = 0.821), and the maximum VAS score was 5 in the ICNB group and 5 in the control group (*p* = 0.589). There were no significant differences in VAS scores between the ICNB and control groups from the operative day to postoperative day seven.

## 4. Discussion

We evaluated the effect of intraoperative ICNBs on postoperative pain after DTI breast reconstruction in cases of NSM or SSM using VAS scores and demonstrated that ICNBs significantly reduced postoperative pain in subpectoral DTI reconstruction during the immediate postoperative period; however, there were no significant differences in VAS scores observed between the ICNB and control groups in pre-pectoral reconstruction. Among patients who underwent subpectoral DTI reconstruction, the average VAS score was significantly decreased on the operative day and the maximum VAS score was significantly decreased on the operative day and postoperative day one in the ICNB group. There were no immediate complications related to the nerve block procedure observed.

Several studies have evaluated the effect of ICNBs on postoperative pain after implant-based breast reconstruction. However, to the best of our knowledge, no studies have examined the differences between subpectoral and pre-pectoral reconstructions. Butz et al. [22] evaluated the effect of intraoperative injection of liposomal bupivacaine in immediate breast reconstruction with the subpectoral placement of a tissue expander and demonstrated that the length of hospital stay and postoperative VAS pain scores were significantly lower in the liposomal bupivacaine group as compared with the pain pump and control groups. On the one hand, Shah et al. [1] assessed the effect of intraoperative administration of bupivacaine in subpectoral DTI reconstruction and found that the consumption of pain medication and length of hospital stay were significantly decreased among patients receiving ICNBs as compared with those who did not. On the other hand, Lanier et al. [4] compared the quality of recovery scores, pain scores, and opioid consumption between the ICNB and placebo groups for patients undergoing immediate breast reconstruction with subpectoral placement of a tissue expander and observed no significant differences between the two groups.

Intercostal nerves arise from the anterior divisions of the thoracic spinal nerves from T1 to T11 [23]. Intercostal nerves supply the sensory innervation for the back, trunk, and upper abdomen as well as the muscular innervation for the intercostal muscles. In addition to distribution to the muscle and skin, branches of the intercostal nerves supply the parietal pleura, mammary glands, and periosteum of the ribs [23,24]. The lateral cutaneous branches are derived from the intercostal nerves around midway between the vertebrae and sternum, which give cutaneous information from the skin of the lateral thoracic wall. The anterior cutaneous branches are the terminal branches of the intercostal nerves, which supply the skin of the anterior thoracic wall. In this study, the exact reason for the superior pain control using ICNBs in subpectoral reconstruction as compared with pre-pectoral reconstruction remains unclear; however, we hypothesized that sensory block of the periosteum could have contributed to the increased effectiveness of ICNBs in subpectoral reconstruction as compared with pre-pectoral reconstruction. The periosteum is highly vascularized and highly innervated by both sympathetic and pain-sensitive fibers [25,26], and mechanical destruction or distortion can cause significant pain [27]. ICNB has been demonstrated to be effective in the management of bone pain [28], which was derived from damage of the periosteum [29]. In subpectoral reconstruction, the periosteum can be easily damaged during the dissection of the subpectoral space and hemostasis of the well-vascularized periosteum. In addition, direct compression of the periosteum by the subpectoral placement of the implant can irritate or distort the periosteum and cause pain. The pain derived from the periosteum of the ribs might be managed by ICNBs. To block collateral branches to the periosteum, the nerve trunk should be targeted rather than cutaneous branches when performing ICNBs.

The findings of previous studies support our hypothesis that sensory block of the periosteum can contribute to the effectiveness of ICNBs in subpectoral breast reconstruction. A study by Lanier et al. [4] revealed that intraoperative nerve blocks failed to improve pain scores in subpectoral tissue expander reconstruction. They performed nerve blocks by injecting bupivacaine and dexamethasone around the anterior and lateral cutaneous branches of the intercostal nerves. In addition, Shah et al. [1] targeted the intercostal nerve trunk in their case series involving subpectoral DTI breast reconstruction and showed a significant decrease in the consumption of pain medication and length of hospital stay following ICNBs. We believe that sensory block of the periosteum by targeting the intercostal nerve trunk is an essential part of ICNBs in subpectoral breast reconstruction.

When VAS scores were compared among patients who did not receive ICNBs, scores in the immediate postoperative period were lower in the pre-pectoral reconstruction group than in the subpectoral reconstruction group. Although the difference did not reach statistical significance, the average VAS score on the operative day was lower in the pre-pectoral reconstruction group with borderline significance (6 versus 7, *p* = 0.062). Previous studies have shown controversial results; some studies reported significantly lower pain scores in pre-pectoral reconstruction than in subpectoral reconstruction [30,31,32], whereas other studies reported that the pain scores were not significantly different [33,34]. Further meta-analysis should be conducted to confirm the significantly lower postoperative pain in pre-pectoral reconstruction as compared with subpectoral reconstruction.

ICNBs could be recommended for patients undergoing subpectoral DTI breast reconstruction for three reasons. First, according to the results of this study, a rapid reduction in VAS pain scores was observed with each successive postoperative day in the immediate postoperative period after subpectoral DTI breast reconstruction. The medians of average VAS scores were 7 (5–8), 5 (3.25–6.75), and 4 (3–5) on the operative day, postoperative day one, and postoperative day two, respectively, among patients who did not receive ICNBs. ICNBs can reduce immediate postoperative pain, and pain after the immediate postoperative period can be effectively managed with conventional analgesia. Secondly, ICNBs have minimal risk for procedure-related complications. Complications after ICNBs include pneumothorax, systemic toxic reactions to local anesthetics, abscess formation, and neuritis [10]. Among them, pneumothorax can be the most severe complication; however, its incidence after ICNBs is rare and has been reported as 0.073–4% [35,36,37]. Moore et al. [35] reported that therapeutic intervention was not required for pneumothorax in their analysis of 10,941 ICNB cases, and severe systemic toxic reactions did not occur. Third, ICNBs can be easily performed by the operating surgeon and do not require an anesthesiologist and the additional positioning of the patient, whereas paravertebral blocks are administered preoperatively by an anesthesiologist while the patient is awake, which can cause significant discomfort to the patient [1]. Moreover, paravertebral blocks bear certain risks according to the neuroaxial location [38]. In terms of the efficacy of the nerve block procedures, ICNB has been demonstrated to be effective in breast surgeries and combined blockade of the pectoral nerves, the intercostobrachial, intercostals III–IV–V–VI, and the long thoracic nerve (Pecs II block) was suggested to be more effective than paravertebral blocks in breast surgeries [39,40]. Nerve block targets for effective pain control in breast surgeries could be a good subject for further research.

The current study had some limitations. First, in the subgroup analysis, the average postoperative VAS score was higher in the subpectoral reconstruction group than in the pre-pectoral reconstruction group for patients who did not receive ICNBs at early postoperative days; however, the difference did not reach statistical significance. A possible explanation for the lack of statistical significance is the small number of patients. Further large-scale studies would be necessary to compare the postoperative pain scores between the subpectoral and pre-pectoral reconstruction groups. Secondly, the daily dose of pain medication could not be assessed because the remaining dose of IV PCA was measured after the IV PCA device was removed. The daily dose of pain medication could be higher in the subpectoral reconstruction than in the pre-pectoral reconstruction because the average and maximum VAS scores were significantly higher in the subpectoral reconstruction in the immediate postoperative period, but the difference of the daily dose of pain medication was not assessed. Last, the groups could be skewed due to the retrospective study design. Taking into consideration the results of this study, a double-blind randomized clinical trial should be performed to confirm the results of this study.

## 5. Conclusions

ICNBs could be a safe and effective method for the pain control of patients undergoing subpectoral DTI breast reconstruction. Intraoperative ICNBs can effectively reduce immediate postoperative pain in subpectoral DTI breast reconstruction; however, it may not be effective in pre-pectoral DTI reconstruction. The nerve trunk should be targeted to block collateral branches to the periosteum of the ribs when performing ICNBs.

## Figures and Tables

**Figure 1 medicina-56-00325-f001:**
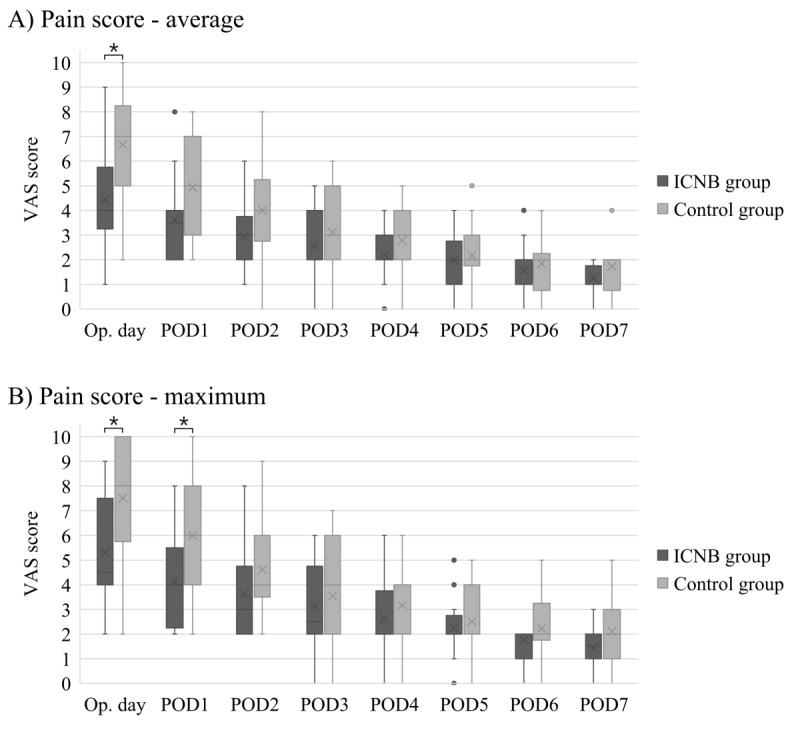
Comparison of postoperative daily average (**A**) and maximum (**B**) visual analogue scale (VAS) pain scores between the intercostal nerve block (ICNB) and control groups for patients who underwent subpectoral direct-to-implant (DTI) breast reconstruction. POD, postoperative day. * means that the differences are statistically significant; • indicates outliers in the box-and-whisker plots.

**Figure 2 medicina-56-00325-f002:**
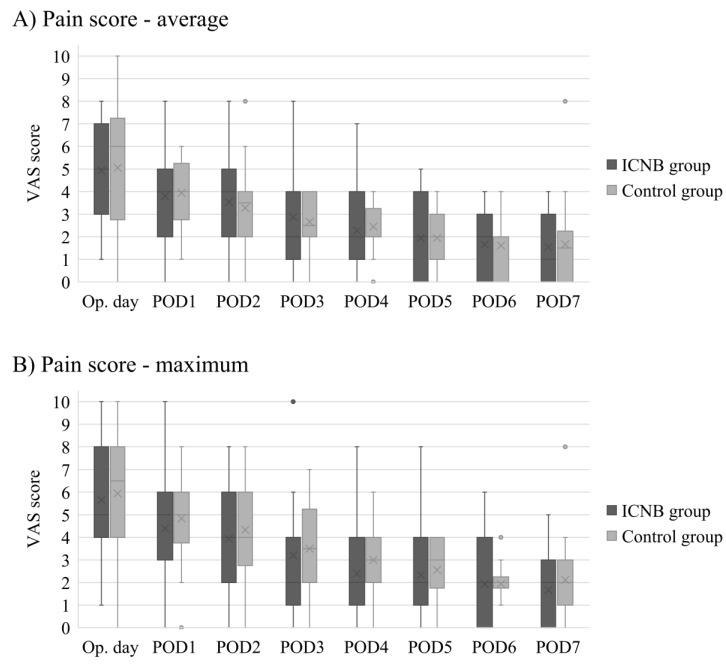
Comparison of postoperative daily average (**A**) and maximum (**B**) VAS pain scores between the ICNB and control groups for patients who underwent pre-pectoral DTI breast reconstruction. • indicates outliers in the box-and-whisker plots.

**Table 1 medicina-56-00325-t001:** Clinical and surgical characteristics.

	Overall Patients (n = 67)	Pre-Pectoral (n = 33)	Subpectoral (n = 34)	*p*
Age, mean ± SD, yr	47.91 ± 8.06	47.12 ± 8.37	48.68 ± 7.80	0.434 *
BMI, mean ± SD, kg/m^2^	22.63 ± 2.63	22.34 ± 2.63	22.92 ± 2.64	0.377 *
Charlson comorbidity index, median (IQR), score	2 (2–3)	2 (2–3)	2 (2–3)	0.911 ^†^
Smoking history, n (%)	3 (4.1)	3 (9.1)	0	0.114 ^§^
Preoperative chemotherapy, n (%)	6 (9.0)	3 (9.1)	3 (8.8)	>0.999 ^§^
Mastectomy type, n (%)				0.673 ^§^
Nipple-sparing	61 (91.0)	31 (93.9)	30 (88.2)	
Skin-sparing	6 (9.0)	2 (6.1)	4 (11.8)	
Mastectomy weight, mean ± SD, g	274.9 ± 133.6	236.4 ± 100.7	312.3 ± 151.5	0.019 *
Implant volume, mean ± SD, mL	306.1 ± 99.8	285.9 ± 85.0	325.7 ± 110.0	0.103 *
Axillary lymph node management, n (%)				0.701 ^‡^
None	0	0	0	
SLNB	56 (83.6)	27 (81.8)	29 (85.3)	
ALND	11 (16.4)	6 (18.2)	5 (14.7)	
Morphine equivalents, median (IQR), mg				
PACU	3 (0–5)	5 (3–5)	3 (0–5)	0.474 ^†^
After the PACU	189.5 (184.5–198.75)	189.5 (184.5–198)	191.5 (184.5–199)	0.339 ^†^

SD, standard deviation; BMI, body mass index; IQR, interquartile range; SLNB, sentinel lymph node biopsy; ALND, axillary lymph node dissection; PACU, post-anesthesia care unit. * *p* value obtained in 2-sampled *t*-test. ^†^
*p* values obtained in Mann–Whitney U test. ^‡^
*p* value obtained in Chi-squared test. ^§^
*p* values obtained in Fisher’s exact test.

**Table 2 medicina-56-00325-t002:** Comparison of postoperative visual analogue scale (VAS) scores between pre-pectoral and subpectoral reconstructions for patients who did not undergo intercostal nerve block.

	Pre-Pectoral Reconstruction	Subpectoral Reconstruction	*p*
Average VAS, median (IQR), score			
Operative day	6 (3–6.75)	7 (5–8)	0.062 *
POD 1	4 (3–5)	5 (3.25–6.75)	0.122 ^†^
POD 2	3.5 (2–4)	4 (3–5)	0.268 ^†^
POD 3	2.5 (2–4)	3 (2–4.75)	0.448 ^†^
POD 4	2 (2–3)	3 (2–4)	0.293 ^†^
POD 5	2 (1–2.75)	2 (2–3)	0.623 ^†^
POD 6	2 (0.25–2)	2 (1–2)	0.656 ^†^
POD 7	1.5 (0–2)	2 (1–2)	0.571 ^†^
Maximum VAS, median (IQR), score			
Operative day	6.5 (4–8)	8 (6–10)	0.108 ^†^
POD 1	5 (4–6)	6 (4–8)	0.114 ^†^
POD 2	4 (3–6)	4 (4–6)	0.692 ^†^
POD 3	3.5 (2–4.75)	4 (2–5.75)	0.937 ^†^
POD 4	3 (2–4)	4 (2–4)	0.660 ^†^
POD 5	3 (2–4)	2 (2–3.75)	0.840 ^†^
POD 6	2 (2–2)	2 (2–3)	0.577 ^†^
POD 7	2 (1–2.75)	2 (1.25–3)	0.720 ^†^

VAS, visual analogue scale; IQR, interquartile range; POD, postoperative day. * *p* value obtained in 2-sampled *t*-test. ^†^
*p* values obtained in Mann–Whitney U test.

**Table 3 medicina-56-00325-t003:** Comparison of clinical and surgical characteristics between the intercostal nerve block (ICNB) and control groups in overall patients.

	ICNB Group (n = 31)	Control Group (n = 36)	*p*
Age, mean ± SD, yr	48.26 ± 7.81	47.61 ± 8.37	0.746 *
BMI, mean ± SD, kg/m^2^	22.66 ± 2.63	22.62 ± 2.67	0.953 *
Charlson comorbidity index, median (IQR), score	2 (2–3)	2 (2–3)	0.928 ^†^
Smoking history, n (%)	1 (3.2)	2 (5.6)	>0.999 ^§^
Preoperative chemotherapy, n (%)	3 (9.7)	3 (8.3)	>0.999 ^§^
Mastectomy type, n (%)			0.404 ^§^
Nipple-sparing	27 (87.1)	34 (94.4)	
Skin-sparing	4 (12.9)	2 (5.6)	
Mastectomy weight, mean ± SD, g	251.6 ± 124.1	295.0 ± 139.9	0.187 *
Implant volume, mean ± SD, mL	282.6 ± 81.7	326.4 ± 110.2	0.073 *
Axillary lymph node management, n (%)			0.953 ^‡^
None	0	0	
SLNB	26 (83.9)	30 (83.3)	
ALND	5 (16.1)	6 (16.7)	
Morphine equivalents, median (IQR), mg			
PACU	5 (3–5)	3 (0–5)	0.474 ^†^
After the PACU	189.5 (180.5–198.75)	191.5 (184.95–198)	0.339 ^†^

ICNB, intercostal nerve block; SD, standard deviation; BMI, body mass index; IQR, interquartile range; SLNB, sentinel lymph node biopsy; ALND, axillary lymph node dissection; PACU, post-anesthesia care unit. * *p* value obtained in 2-sampled *t*-test. ^†^
*p* values obtained in Mann–Whitney U test. ^‡^
*p* value obtained in Chi-squared test. ^§^
*p* values obtained in Fisher’s exact test.

**Table 4 medicina-56-00325-t004:** Comparison of postoperative VAS scores between the ICNB and control groups in overall patients.

	ICNB Group	Control Group	*p*
Average VAS, median (IQR), score			
Operative day	4 (3.5–6)	6 (4–8)	0.047 *
POD 1	4 (2–4.5)	4 (3–6)	0.104 ^†^
POD 2	3 (2–4)	4 (2–4.25)	0.365 *
POD 3	2 (1–4)	3 (2–4)	0.487 ^†^
POD 4	2 (1–3)	2.5 (2–4)	0.108 ^†^
POD 5	2 (1–3)	2 (1–3)	0.688 ^†^
POD 6	1 (1–2)	2 (0.75–2)	0.190 ^†^
POD 7	1 (0.5–2)	2 (0–2)	0.117 ^†^
Maximum VAS, median (IQR), score			
Operative day	5 (4–8)	7.5 (5–8)	0.030 ^†^
POD 1	4 (3–5.5)	6 (4–7.25)	0.030 *
POD 2	4 (2–4.5)	4 (3–6)	0.137 ^†^
POD 3	2 (2–4)	4 (2–5.25)	0.301 ^†^
POD 4	2 (2–3.5)	3 (2–4)	0.068 ^†^
POD 5	2 (1–3)	2 (2–4)	0.272 ^†^
POD 6	2 (1–2)	2 (2–3)	0.190 ^†^
POD 7	2 (0.5–2)	2 (1–3)	0.117 ^†^

ICNB, intercostal nerve block; VAS, visual analogue scale; IQR, interquartile range. * *p* value obtained in 2-sampled *t*-test. ^†^
*p* values obtained in Mann–Whitney U test.

**Table 5 medicina-56-00325-t005:** Comparison of clinical and surgical characteristics between the ICNB and control groups for patients who underwent subpectoral reconstruction.

	ICNB Group (n = 16)	Control Group (n = 18)	*p*
Age, mean ± SD, yr	46.69 ± 7.56	47.78 ± 8.11	0.484 *
BMI, mean ± SD, kg/m^2^	23.1 ± 2.56	22.75 ± 2.77	0.696 *
Charlson comorbidity index, median (IQR), score	2 (2–4)	2 (2e3)	0.874 ^†^
Smoking history, n (%)	0	0	-
Preoperative chemotherapy, n (%)	2 (12.5)	1 (5.6)	0.591 ^‡^
Mastectomy type, n (%)			>0.999 ^‡^
Nipple-sparing	15 (93.8)	16 (88.9)	
Skin-sparing	1 (6.3)	2 (11.1)	
Mastectomy weight, mean ± SD, g	277.9 ± 136.8	342.9 ± 160.9	0.217 *
Implant volume, mean ± SD, mL	292.2 ± 92.0	355.6 ± 118.4	0.094 *
Axillary lymph node management, n (%)			>0.999 ^‡^
None	0	0	
SLNB	14 (87.5)	15 (83.3)	
ALND	2 (12.5)	3 (16.7)	
Morphine equivalents, median (IQR), mg			
PACU	3 (0–5)	4 (0–5)	0.880 ^†^
After the PACU	189.5 (181–196.5)	191.5 (185.75–204.3)	0.115 ^†^

ICNB, intercostal nerve block; SD, standard deviation; BMI, body mass index; IQR, interquartile range; SLNB, sentinel lymph node biopsy; ALND, axillary lymph node dissection; PACU, post-anesthesia care unit. * *p* value obtained in 2-sampled *t*-test. ^†^
*p* values obtained in Mann–Whitney U test. ^‡^
*p* values obtained in Fisher’s exact test.

**Table 6 medicina-56-00325-t006:** Comparison of clinical and surgical characteristics between ICNB and control groups for patients who underwent pre-pectoral reconstruction.

	ICNB Group(n = 15)	Control Group(n = 18)	*p*
Age, mean ± SD, yr	46.73 ± 8.04	47.44 ± 8.85	0.812 *
BMI, mean ± SD, kg/m^2^	22.17 ± 2.70	22.49 ± 2.65	0.738 *
Charlson comorbidity index score, median (IQR), score	2 (2–3)	2 (2–3.75)	0.997 ^†^
Smoking history, n (%)	1 (6.7)	2 (11.1)	>0.999 ^‡^
Preoperative chemotherapy, n (%)	1 (6.7)	2 (11.1)	>0.999 ^‡^
Mastectomy type, n (%)			>0.999 ^‡^
Nipple-sparing	13 (86.7)	18 (100.0)	
Skin-sparing	2 (13.3)	0	
Mastectomy weight, mean ± SD, g	223.5 ± 106.2	247.1 ± 97.7	0.532 *
Implant volume, mean ± SD, mL	272.3 ± 70.8	297.2 ± 95.8	0.411 *
Axillary lymph node management, n (%)			>0.999 ^‡^
None	0	0	
SLNB	12 (80.0)	15 (83.3)	
ALND	3 (20.0)	3 (16.7)	
Morphine equivalents, median (IQR), mg			
PACU	3 (0–5)	5 (3–5)	0.901 ^†^
After the PACU	189.5 (185.9–202)	189.5 (184.65–196)	0.714 ^†^

ICNB, intercostal nerve block; SD, standard deviation; BMI, body mass index; IQR, interquartile range; SLNB, sentinel lymph node biopsy; ALND, axillary lymph node dissection; PACU, post-anesthesia care unit. * *p* value obtained in 2-sampled *t*-test. ^†^
*p* values obtained in Mann–Whitney U test. ^‡^
*p* values obtained in Fisher’s exact test.

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
