# Peer review of "Intraoperative Intercostal Nerve Block for Postoperative Pain Control in Pre-Pectoral versus Subpectoral Direct-To-Implant Breast Reconstruction: A Retrospective Study"

_medicina, 2020, doi:10.3390/medicina56070325_

Round 1

Reviewer 1 Report

Upon detailed review of the manuscript, these are my comments

  1. This is a  randomized controlled trial comparing the pain scores between groups who received intercostal nerve blocks and control and between two different techniques of breast reconstruction. The study is designed in such a way that pain scores and morphine requirements are all measured after the procedure in the PACU. It should be noted that the analgesics given during the procedures will have an impact in the postoperative period at least for several hours in the recovery room. This was not mentioned anywhere in the manuscript whether there is a standardization of the anesthetic management between the study subjects and the controls. And rather not mentioned anywhere what type of anesthetic was administered. 
  2. It is shown that  Intercostal nerve blocks are effective in the immediate postoperative period in the subpectoral group and not in the pre pectoral group. But there is no difference noted in the morphine equivalent requirements between the two.  Though VAS scores are lower in the immediate postoperative period, pain scores are never considered a surrogate for narcotic requirements for the study design like this. 
  3. Lines 102-105:  Fentanyl PCA was used for 48 hours for postoperative pain and non-opioid pain medications were administered as needed after 48 hours. It was not mentioned on what postoperative day the patient was discharged. Were the patients admitted for a week after the surgery?  Early mobilization and discharge from the hospital are important indicators for good pain control and decreased opioid requirements. After the patients were discharged, how did they instruct the patients to record pain scores and medications received. And how did they follow up with the patients to enquire about pain scores?
  4. VAS scores are analyzed in the statistical section as mean +SD, usually, they should be measured in the median with interquartile range. Nevertheless, VAS scores differences are not good indicators of postoperative pain, as it is a subjective variable. VAS score of 8 should not be considered as double the pain compared to the VAS score of 4. 
  5. 255-264- The hypothesis for better pain control in the subpectoral group who received ICNBs compared to pre pectoral group seems very controversial. Intercostal nerve block provides analgesia to the skin and muscles of the anterior chest wall apart from the periosteum, which was discussed in the discussion section about the intercostal nerves. If it provides analgesia in cases where periosteum is damaged, then it should provide better analgesia for surgeries which does not damage the periosteum with a similar incision size and site. 

Author Response

Firstly, I deeply appreciate your insightful comments to this manuscript entitled, “Intraoperative Intercostal Nerve Block for Postoperative Pain Control in Prepectoral versus Subpectoral Direct-to-Implant Breast Reconstruction”. I very much feel your comments have helped us polish our manuscript. I have attached my revisions in addition to an itemized, point-by-point response to your comments.

1. This is a randomized controlled trial comparing the pain scores between groups who received intercostal nerve blocks and control and between two different techniques of breast reconstruction. The study is designed in such a way that pain scores and morphine requirements are all measured after the procedure in the PACU. It should be noted that the analgesics given during the procedures will have an impact in the postoperative period at least for several hours in the recovery room. This was not mentioned anywhere in the manuscript whether there is a standardization of the anesthetic management between the study subjects and the controls. And rather not mentioned anywhere what type of anesthetic was administered.

Response: The anesthetic management in the PACU was not standardized in this study, so we compared the dose of anesthetics administered in the PACU between the study and control groups. “After the surgery, fentanyl was administered intravenously according to patient’s vital signs or complain of pain in the post-anesthesia care unit (PACU).” (lines 99-100, based on the state that the track changes function in Microsoft Word is activated) and “Morphine equivalents administered in the PACU and after the PACU were analyzed.” (lines 110-111) were added to the 3rd paragraph of the Materials and Methods section. Pain medications used in the PACU were analyzed and added to the table 1, 3, 5, and 6. In addition, we routinely administered IV fentanyl with a dose of 3 µg per kg of the patient’s body weight during the surgery. “Under the general anesthesia, IV fentanyl (Hana Pharm Co., Ltd., Seoul, Republic of Korea) was routinely administered with a dose of 3 µg per kg of the patient’s body weight.” was added to the 2nd paragraph of the Materials and Methods section (lines 79-80).

2. It is shown that Intercostal nerve blocks are effective in the immediate postoperative period in the subpectoral group and not in the pre pectoral group. But there is no difference noted in the morphine equivalent requirements between the two.  Though VAS scores are lower in the immediate postoperative period, pain scores are never considered a surrogate for narcotic requirements for the study design like this.

Response: We did not used the VAS scores as a surrogate for narcotic requirements, but collected VAS score with a uniform manner by using same reporting sheet and patient education regarding criteria of VAS scores (4th paragraph of Materials and Methods section). On the other hands, narcotic requirements might not reflect degree of pain. Because IV PCA was used in all patients, baseline infusion rates of opioids were same in all patients. Patients were encouraged to press the button of IV PCA device for additional input of the opioids by the PCA when they feel pain. “Patients were encouraged to press the button whenever they feel pain.” was added to the 3rd paragraph of the Materials and Methods section (line 105) and two sentences were added to the last paragraph of Discussion section, indicating lack of data regarding daily dose of pain medication is a limitation of this study (lines 327-332).

3. Lines 102-105:  Fentanyl PCA was used for 48 hours for postoperative pain and non-opioid pain medications were administered as needed after 48 hours. It was not mentioned on what postoperative day the patient was discharged. Were the patients admitted for a week after the surgery?  Early mobilization and discharge from the hospital are important indicators for good pain control and decreased opioid requirements. After the patients were discharged, how did they instruct the patients to record pain scores and medications received. And how did they follow up with the patients to enquire about pain scores?

Response: According to your comments, “Patients were discharged after all of surgical drains removed.” was added to the 2nd paragraph of the Materials and Methods section (lines 96-97) and “If patients were discharged earlier than postoperative day 7, oral pain medication was prescribed and patients were asked to fulfill the self-recording VAS score sheet and the sheets were retrieved at the first visit to the outpatient clinic. All patients except two discharged after 7 postoperative days.” to the 3rd paragraph (lines 118-121). Among 67 patients included in this study, only two patients were discharged earlier than postoperative day 7.

4. VAS scores are analyzed in the statistical section as mean +SD, usually, they should be measured in the median with interquartile range. Nevertheless, VAS scores differences are not good indicators of postoperative pain, as it is a subjective variable. VAS score of 8 should not be considered as double the pain compared to the VAS score of 4.

Response: According to your recommendation, mean ± SD has been changed to median (interquartile range) to present VAS scores in the abstract (lines 23-28), results section (lines 160-161, Table 2, lines 177-178, Table 4, lines 198-203, and lines 222-226) and discussion section (line 295 and lines 303-304). Thank you for your insightful suggestion and we feel the manuscript has benefited significantly from your comments. We agree with your point that VAS score is not an objective measurement of the pain. Therefore, we educated all of the patients the day before the surgery when the patient received a reporting sheet of VAS score. We educated that a VAS score of 5 was a degree of pain at which the patient found it difficult to sleep or rest without additional pain control (lines 114-116). We believe this education and using a uniform reporting sheet could lead to positive results in this study.

5. 255-264- The hypothesis for better pain control in the subpectoral group who received ICNBs compared to pre pectoral group seems very controversial. Intercostal nerve block provides analgesia to the skin and muscles of the anterior chest wall apart from the periosteum, which was discussed in the discussion section about the intercostal nerves. If it provides analgesia in cases where periosteum is damaged, then it should provide better analgesia for surgeries which does not damage the periosteum with a similar incision size and site.

Response: We agree with that the hypothesis is controversial and not yet well studied. However, ICNB has been demonstrated to be effective in bone pain and postoperative pain after thoracotomy surgeries and we think the hypothesis need to be studied further. We changed the word “believe” into “hypothesized in line 270 and “can” to “might” in line 279 (3rd paragraph of the Discussion section) and added a sentence supporting our hypothesis in the lines 274-275.

Once again, we extend thanks and appreciation to your reviews and valuable comments on our manuscript.

Sincerely yours,

Kyong-Je Woo

Reviewer 2 Report

In this study the main problem is the methods section 

the reader cannot understand was the study a retrospective or prospective 

if there was a prospective study , there is no clear explanation why  some patients had the block and some did'nt the authors should describe extensively in which criteria the block was administered and write the exact nature of the study fron this  text I believe it is a retropctive 

Results should be better described 

Pain score are better represented by Box plot 

The authors should also discuss analgesia provided by paravertebral block or pecs block  in this type of surgery which are also new techniques for postoperative pain relief

Author Response

Firstly, I deeply appreciate your insightful comments to this manuscript entitled, “Intraoperative Intercostal Nerve Block for Postoperative Pain Control in Prepectoral versus Subpectoral Direct-to-Implant Breast Reconstruction”. I very much feel your comments have helped us polish our manuscript. I have attached my revisions in addition to an itemized, point-by-point response to your comments.

In this study the main problem is the methods section

the reader cannot understand was the study a retrospective or prospective

if there was a prospective study , there is no clear explanation why  some patients had the block and some did'nt the authors should describe extensively in which criteria the block was administered and write the exact nature of the study fron this  text I believe it is a retropctive

Response: This study is a retrospective study. We removed “randomly” from the abstract (line 16, based on the state that the track changes function in Microsoft Word is activated) and “Prospectively recorded data from” (line 65) and “Patients who received ICNBs were selected using computer-generated random numbers.” (lines 74-75) from the first paragraph of the Materials and Methods section to prevent confusion regarding the nature of this study.

Results should be better described

Pain score are better represented by Box plot

Response: According to your recommendation, figures 1 and 2 has been changed to Box plots.

The authors should also discuss analgesia provided by paravertebral block or pecs block  in this type of surgery which are also new techniques for postoperative pain relief

Response: According to your recommendation, we added three sentences to the 6th paragraph of Discussion section to expand the discussion regarding paravertebral block and to include discussion regarding pecs block. (lines 316-321).

Once again, we extend thanks and appreciation to your reviews and valuable comments on our manuscript.

Sincerely yours,

Kyong-Je Woo

Round 2

Reviewer 1 Report

Thank you for revising the manuscript as per the suggestions from the reviews.

Upon reviewing the revised manuscript on page 10, table 6, morphine equivalents in ICNB and control group is deleted. In studies comparing postoperative pain management, opioid consumption is an important variable to compare. Even though the difference is not statistically significant, but that an important variable.  

Author Response

Upon reviewing the revised manuscript on page 10, table 6, morphine equivalents in ICNB and control group is deleted. In studies comparing postoperative pain management, opioid consumption is an important variable to compare. Even though the difference is not statistically significant, but that an important variable.

Response: Thank you for your review on this manuscript. Morphine equivalents data was removed from the original position, but added to lower lines to present morphine equivalents both in the PACU and after the PACU in Table 6 (lines 225-230). Please let us know if we misunderstood your intended meaning.

Reviewer 2 Report

Since the nature of the manuscript is now clear I would rather add the end of the title : A retrospective study

a final suggestion toward prospective randomized study should be made since this results can be used only  a prelude to be confirmed further

otherwise the authors has responded adequatly to most of the questions

Author Response

Since the nature of the manuscript is now clear I would rather add the end of the title : A retrospective study

a final suggestion toward prospective randomized study should be made since this results can be used only  a prelude to be confirmed further

otherwise the authors has responded adequatly to most of the questions

Response: Thank you for your review on this manuscript. According to your recommendation, “: A Retrospective Study” was added to the end of the title (lines 4-5). The last sentence of Discussion section was modified to emphasize prospective randomized study should be performed to confirm the results of this study (lines 329-330).